# Electrospinning of Microstructures Incorporated with Vitamin B9 for Food Application: Characteristics and Bioactivities

**DOI:** 10.3390/polym14204337

**Published:** 2022-10-15

**Authors:** Sílvia Castro Coelho, Fernando Rocha, Berta Nogueiro Estevinho

**Affiliations:** 1LEPABE, Departamento de Engenharia Química, Faculdade de Engenharia da Universidade do Porto, Rua Dr. Roberto Frias, 4200-465 Porto, Portugal; 2ALiCE—Associate Laboratory in Chemical Engineering, Faculty of Engineering, University of Porto, Rua Dr. Roberto Frias, 4200-465 Porto, Portugal

**Keywords:** zein protein, modified starch, vitamin B9, plasticizers, electrospinning gelatinization, antioxidant activity, food applications

## Abstract

The food industry has been expanding, and new vectors to entrap vitamins have been constantly investigated, aiming at versatile systems with good physico-chemical characteristics, low-cost production, high stability and the efficient release of active ingredients. The vitamin B9 (folic acid or folate) is essential for the healthy functioning of a variety of physiological processes in humans and is beneficial in preventing a range of disorders. In this study, two approaches were developed to encapsulate vitamin B9. Zein and the combination of modified starch with two plasticizers were the selected encapsulating agents to produce microstructures via the electrospinning technique. The objective was to improve the stability and the B9 antioxidant capacity in the final formulations. The work strategy was to avoid limitations such as low bioavailability, stability and thermosensitivity. The microstructures were fabricated and the morphology and shape were assessed by scanning electron microscopy. The B9 release profiles of modified starch and zein microstructures were analyzed in simulated gastric fluid at 37 °C, and in deionized water and ethanol at room temperature. The B9 encapsulation efficiency and the stability of the systems were also studied. The ABTS assay was assessed and the antioxidant activity of the produced microstructures was evaluated. The physico-chemical characterization of loaded B9 in the microstructures was achieved. High encapsulation efficiency values were achieved for the 1% B9 loaded in 12% *w*/*w* modified starch film; 5% B9 vitamin encapsulated by the 15% *w*/*w* modified starch with 4% *w*/*w* tween 80; and 4% *w*/*w* glycerol film with heterogeneous microstructures, 5% *w*/*w* zein compact film and 10% *w*/*w* zein film. In conclusion, the combinations of 7 wt.% of modified starch with 4 wt.% tween 80 and 4 wt.% glycerol; 15 wt.% of modified starch with 4 wt.% tween 80 and 4 wt.% glycerol; and 12 wt.% modified starch and 5 wt.% zein can be used as delivery structures in order to enhance the vitamin B9 antioxidant activity in the food and nutraceutical fields.

## 1. Introduction

Electrospun/electrosprayed microstructures are systems that are able to overcome the limitations associated with the presented vitamins in functional food and nutraceutical products [1,2]. The low bioavailability, stability and half-life time are the main drawbacks of these ingredients [3]. These vectors present physico-chemical properties such as high porosity and large surface-to-volume and good mechanical characteristics (flexibility) that might improve the vitamin’s stability and enhance its protection and bioavailability to be applied in functional food and nutraceutical products [4].

Electrospinning and electrospraying are efficient electrohydrodynamic techniques to produce low-cost and versatile polymeric microstructures (fibers, films and/or particles) [5]. The structures’ walls entrap the biocompounds, allowing their preservation and stability [6]. 

The application of an electric field to the polymeric solution generates the formation of a Taylor cone. The charged polymer is emanated and solidified to form the structures that will be deposited on a grounded collector [7,8]. The major difference between these techniques is the polymeric solution viscosity, which allows the continuous ejection of liquid to produce fibers or its breakage into droplets (particles), respectively [8]. Processing parameters (voltage, distance, flow rate), polymeric characteristics (surface tension, electrical conductivity, molecular weight) and environmental conditions (temperature, humidity) have an influence on the size, morphology, structure and porosity of the fabricated systems [9].

Vitamin B9 [(2*S*)-2-[[4-[(2-amino-4-oxo-1*H*-pteridine-6-yl) methylamino] benzoyl]amino] pentanedioic acid] is the selected ingredient in this work. Known also as folic acid (sintetic form) or folate (natural form), it is a water-soluble micronutrient that presents an essential role in the process of mitotically active tissues, as well as in DNA synthesis and repair [10,11]. 

Folic acid is given to women at the beginning of pregnancy to minimize the risk of neural tube defects [12,13]. It has been used in food product fortification, tissue engineering development and cancer prevention, such as colorectal adenomas and their treatment [14,15]. B9 presents associated limitations that must be considered when it is applied. The drawbacks such as low specificity and bioavailability make them suitable compounds to be studied as encapsulated agents [16]. Evangelho et al. reported the high encapsulation of folic acid (in the range of 74.0% and 93.5%) by zein fibers and capsules produced by electrospinning and electrospraying methods, respectively [17]. The developed systems presented good properties, such as high resistance against light (UVA) irradiation as well as against the thermal conditions (100, 140 and 180 °C) used for food applications. Fonseca et al. suggested starch fibers as systems for the high encapsulation efficiency of folic acid that might be used in the food industry [18]. Panalva et al. prepared successfully zein nanoparticles with folic acid to improve oral bioavailability [19]. 

The objective of this study was the encapsulation of vitamin B9 into polymeric microstructures developed by the electrospinning method. Modified starch (MS) and zein were the chosen biocompatible polymers. Both natural polymers present promising characteristics, such as biocompatibility, biodegradability, non-toxicity and low cost [6,20]. Moreover, the influence of plasticizers—tween 80 and glycerol—on the gelatinization of the modified starch and, therefore, on the developed crystalline structures was investigated. Crochet et al. reported the influence of sucrose on the solubility and gelatinization of starch [21]. Gamarano et al. presented modifications of the crystal structures of starch when glycerol and urea were incorporated into the system. The results showed significative changes in the matrices’ morphology, which might increase the fertilizers’ release. Niu et al. suggested the production of zein-ethyl cellulose electrospun fibers for food preservation [22]. It was observed that the vectors were able to prolong the shelf-life time of *Agaricus bisporus* mushroom. Zein electrospun fibers were produced by Ansarifar et al. to preserve strawberry [23]. The system loaded an essential oil and the results suggested shelf-life extension and fruit quality preservation during the storage process. Sateike et al. reported the optimization of PVA/modified starch nanofibers produced by the electrospinning technique based on the morphology and structure of the developed systems [24]. Aiming at the biocompounds’ protection, Coelho et al. investigated zein microstructure fabrication [25]. 

In this context, electrospun/electrosprayed microstructures were developed and characterized by the scanning electron microscopy (SEM) technique, in terms of in vitro drug release, in vitro vitamin B9 stability and antioxidant activity. The effect of using plasticizers on modified starch behavior was shown for the first time and suggests that it might be responsible for the better stability of vitamin B9 in the developed matrices while preventing the crystalline morphology.

## 2. Materials and Methods

### 2.1. Materials

High-purity and pharmaceutical-grade reagents were selected in this work. Vitamin B9 (folic acid) (MW: 441.40 g/mol; ≥97%) was acquired from Sigma-Aldrich. Zein prolamine (grade Z3625—zein from corn was reported to be approximately 35% α-zein, which includes 2 prominent bands of 22 and 24 kDa) was purchased from Sigma Aldrich (St. Louis, MO, USA) and modified starch ((36673-22)MW: 162.14 g/mol; insolubles 0.01% max) was from Alfa Aesar (Haverhill, MA, USA). Poly(ethylene oxide) (Mv 100,000 (nominal)), tween 80 (P1754-1L, Lot#BCBT8094, CAS 9005-65-6), glycerol (MW: 92.09 g/mol, ≥99.5%), 2.2′-Azino-bis(3-ethylbenzothiazoline-6-sulfonic acid) diammonium salt (MW: 548.68 g/mol) and (±)-6-Hydroxy-2,5,7,8-tetramethylchromane-2-carboxylic acid (Trolox) (MW: 250.29 g/mol) were obtained from Sigma Aldrich (USA). Absolute ethanol (83,813.360, Lot 21B104029, CAS 64-17-5) was acquired from VWR BDH Chemicals.

### 2.2. Preparation of the Zein and Modified Starch Solutions

Zein was dissolved in aqueous ethanol and stirred at room temperature until completely dissolved. Different zein concentrations—5% *w*/*w*, 10% *w*/*w* in 70% (*w*/*w*, weight of solvent/(weight of solvent + weight of water)) ethanol and 30% *w*/*w* in 60% (*w*/*w*, weight of solvent/(weight of solvent + weight of water)) ethanol—were prepared (Table 1).The chosen zein concentrations are in accordance with the preliminary optimization of the method [25].

Modified starch was dissolved with stirring in deionized water at room temperature. Tween 80 and glycerol were added to the dissolved modified starch. The formulations were produced according to Table 1. The optimization of encapsulating agent concentrations as well as electrospinning parameters was performed based on the literature.

Kong et al. reported that 15% *w*/*w* starch fibers can be developed as promising vectors for food and biomedical products [26]. Based on this research and our goal, the optimization of electrospinning parameters began with a 15% *w*/*w* MS concentration. Different concentrations below 15% (7% *w*/*w* and 12% *w*/*w*) were also evaluated, aiming the fabrication of microstructures with small diameters. 

Several reports suggested the combination of plasticizers and modified starch to improve the mechanical characteristics of the structures [27,28]. Therefore, this study had one other goal that consisted of the combination of two plasticizers (glycerol and tween 80) [29,30] with the modified starch in order to enhance the vitamin B9 encapsulation efficiency [31]. 

### 2.3. Production of Microstructures by Electrospinning

The electrospinning experimental set-up (Spraybase^®^ (Dublin, Ireland)), equipped with a variable high-voltage 0–20 kV power supply and a stainless-steel needle 20G, was used to produce the modified starch/zein-based structures, at 22 °C. The effect on the morphology and shape of fibers/beads was investigated. The electrospinning conditions such as the flow rate, the voltage and the tip-to-collector (Ø = 90 mm) distance used are shown in Table 1. 

Vitamin B9 solution with a concentration of 1% *w*/*w* and 5% *w*/*w* was added to the polymer solutions and stirred in the dark.

### 2.4. Characterization of Microstructures

A small amount of sample was fixed on a brass stub using double-sided adhesive tape, followed by coating in a vacuum with a thin layer of gold for 15 min. 

The morphology and shape of structures were assessed in different areas of the samples by scanning electron microscopy (SEM) (Fei Quanta 400 FEG ESEM/EDAX Pegasus X4M, Eindhoven, The Netherlands) at Centro de Materiais da Universidade do Porto (CEMUP), Porto, Portugal. The magnifications used in each sample were 100×, 500×, 1000×, 10,000× and 5000×; the beam intensity (HV) was 15.000 kV.

### 2.5. In Vitro Release Studies

The B9 release profiles of modified starch and zein microstructures were determined for the selected samples in simulated gastric fluid (SGF) at 37 °C, and in deionized water and ethanol at room temperature, respectively.

To validate the method, the calibration curve, coefficient of variation, limit of detection (LOD) and limit of quantification (LOQ) were determined for vitamin B9. Table 2 presents B9’s characteristics. The standard solutions were prepared with SGF fluid (37 °C), deionized water (22 °C) and ethanol (22 °C).

Six standard solutions were prepared with deionized water (22 °C) and SGF (37 °C) within a range of 0.003–0.05 mg/mL; seven standard solutions were prepared with ethanol (22 °C) in the range of 0.0025–0.05 mg/mL. The B9 peak identified and used for this study was 285 ± 1 nm. 

The obtained calibration curve for SGF fluid at 37 °C (y = 32.795x + 0.0185) proved to be linear for this specific range interval, with a good coefficient of correlation (R^2^ = 0.999). The LOD was 0.0013 mg/mL and the LOQ was 0.0041 mg/mL. 

The calibration curve for ethanol (y = 54.537x + 0.0498) was linear in the selected range, with a good coefficient of correlation (R^2^ = 0.998). The LOD was 0.0024 mg/mL and the LOQ was 0.0074 mg/mL. 

The linear calibration curve for deionized water (y = 45.296x + 0.0505) showed a good coefficient of correlation (R^2^ = 0.999); the LOD and LOQ determined were 0.0016 mg/mL and 0.0048 mg/mL, respectively.

The release studies were performed by adding the developed microstructures (3–8 mg) to the release medium (3 mL), incubated with constant magnetic stirring. The B9 concentration of the dialysate buffer was monitored over time, at 285 nm, using a NanoDrop One-C spectrophotometer (Thermo Fisher Scientific, Waltham, MA, USA).

### 2.6. Vitamin Encapsulation Efficiency

The encapsulation efficiency (EE) of vitamin B9 was estimated by measuring the UV light absorbance of the supernatant using a NanoDrop One-C spectrophotometer (Thermo Fisher Scientific, USA). The vitamin in the structures was interpolated from the linear calibration curve. The EE was calculated using Equation (1):(1)$$\mathrm{EE}\;\left(\%\right)=\frac{\mathrm{theoretical}\;\mathrm{vitamin}\;\mathrm{in}\;\mathrm{the}\;\mathrm{structures}\;-\mathrm{supernatant}\;\mathrm{of}\;\mathrm{the}\;\mathrm{sample}}{\mathrm{theoretical}\;\mathrm{vitamin}\;\mathrm{in}\;\mathrm{the}\;\mathrm{structures}}\times 100$$

The samples were centrifuged after production (fresh microstructures) and the UV measurements of the supernatant were analyzed. 

### 2.7. In Vitro Stability Assay

The samples were centrifuged after 2 months to evaluate the stability/loss of vitamin with time. UV measurements of the supernatants were evaluated.

### 2.8. In Vitro Antioxidant Activity Experiment

An ABTS* radical scavenging experiment was used to evaluate the antioxidant activity of the produced microstructures [32]. Succinctly, 7.4 mM ABTS aqueous solution was prepared and mixed with 2.6 mM potassium persulfate aqueous solution, for 16 h in the dark, at room temperature (22 °C). Then, 1 mL of ABTS solution was added to 60 mL of pure ethanol to prepare the ABTS* solution. The samples (150 μL) were mixed with 2850 μL of ABTS* solution and allowed to react for 2 h, at 22 °C and in the dark. The analysis was executed at 734 nm and room temperature using the NanoDrop One-C spectrophotometer (Thermo Fisher Scientific, USA). 

### 2.9. Statistical Analysis

Analytical determinations for the samples were carried out in triplicate and the results were expressed as a mean of the triplicates with standard deviations. The statistical significance results were analyzed (*p* ≤ 0.05) using a single-factor analysis of variance (ANOVA).

## 3. Results

### 3.1. Characterization of Produced Structures in Terms of Morphology and Size and Encapsulation Efficiency (EE)

In the present study, different parameters were optimized in order to produce polymeric structures that might be used to incorporate bioactive ingredients such as vitamin B9, as visualized in Table 1. The prepared microstructures were affected by the modified starch concentration (7 wt.%, 12 wt.% and 15 wt.%). The addition of tween 80/glycerol (4 and 8 wt.% of each) also generated morphologic changes. Preliminary studies were performed to optimize the process characteristics, which are visualized in Table 1. Figure 1 and Figure 2 show SEM images of the unloaded modified starch and zein microstructures prepared by the electrospinning process. 

For modified starch of 7 wt.%, it was found that crystals were formed (Figure 1A). A thin film with spherical structures was generated when tween 80 plus glycerol in the ratio 4 wt.%/4 wt.% and 8 wt.%/8 wt.% was incorporated into the polymer solution (Figure 1B,C). Here, the microstructures presented diameters between 0.76 and 2.43 μm and between 0.40 and 1.64 μm, respectively. This result might be due to the presence of the plasticizers. In fact, the incorporation of glycerol and tween 80 affects the gelatinization and crystalline structure of the polymer, contributing to modified starch hydration [33].

Crystalline structures were visualized on the surface of the 12MS sample. When 4T4G was incorporated into 12MS, porous structures were observed (Figure 1E). In the presence of 8 wt.% tween 80 plus 8 wt.% glycerol (Figure 1F), a film composed of microspheres was visualized. The spheres presented diameters in the range of 2.96 ± 1.29 μm.

For all the prepared matrices with 15MS, crystalline structures were produced (Figure 1G–I).

The microstructures presented in Figure 2 were developed and optimized previously [25]. Figure 2A shows zein microbeads with diameters around 0.74—3.3 μm. An increase in the zein concentration to 10 wt.% (10Z, Figure 2B) generated a homogeneous matrix. With 30Z, smooth electrospun fibers with diameters of approximately 0.37 to 5.28 μm were observed.

Following the optimization of the modified starch and zein microstructures, vitamin B9 was encapsulated using the electrospinning process. The final formulations were evaluated by SEM and characterized in terms of morphology and size. Different conditions were selected according to the optimized parameters in the fabricated structures alone. The microstructures were prepared with 1% and 5% of vitamin B9. Figure 3 shows SEM images of the microstructures produced at 20 kV with 12 wt.% modified starch alone and with the incorporation of tween 80 and glycerol. The observed results suggested that 12MS crystalline structures with 1% B9 were formed, with diameters between 10 and 21 μm. Encapsulating 5% B9 in a similar structure, a thin film was produced (Figure 3B). Moreover, 12MS4T4G microparticles with 1% B9 were formed and can be visualized in Figure 3C. In this situation, the plasticizers might contribute to the structures’ development with different diameters (10.2 ± 2.51 μm) and irregular shapes. Increasing the plasticizers to double the amount, porous matrices were formed (Figure 3E) and a thin film with heterogeneous structures was visualized, as seen in Figure 3F (12MS8T8G-5B9). These results corroborate the literature suggesting that the incorporation of plasticizers—glycerol and tween 80—will improve the tensile strength in the MS, which does not recrystallize quickly [34]. The best structures to use for food applications are produced with 4% *w*/*w* of plasticizer. These circumstances might allow us to achieve better mechanical characteristics in the produced microstructures.

A decrease in the MS concentration to 7% *w*/*w* while maintaining 4T4G indicates the production of microstructures (1% vitamin) and a compact film (5% vitamin) (Figure 4A,B, respectively). On the other hand, an increase in the MS concentration to 15% *w*/*w* allows the fabrication of a compact film and a film with microstructures corresponding to 1% *w*/*w* and 5% *w*/*w* of B9.

In Figure 5, SEM images of zein structures loaded with vitamin B9 (1% and 5%) are presented. The best results were obtained for 5% *w*/*w* zein microstructures. Figure 5A shows small particles produced with diameters in the range of 80 and 435 nm, and in Figure 5B, a compact film can be observed. Similar results were obtained for 10Z.

The encapsulation efficiency of the developed structures was estimated and the results are summarized in Table 3. The highest EE was obtained for 12MS-1B9, 5Z-5B9 and 10Z-5B9. These results suggested that the compact matrices and fibers developed present the best conditions to incorporate this type of vitamin for these specific optimized parameters. Almost all formulations presented good stability of B9 after 2 months of production.

### 3.2. In Vitro Release Experiments

The MS and zein structures were studied in terms of the B9 release profile. This is an important experiment to verify the vitamin B9 behavior in each sample with time, at specific conditions. The release profiles of the selected formulations are depicted in Figure 6, Figure 7, Figure 8 and Figure 9. Considering the release profiles observed in Figure 6, the encapsulation of vitamin B9 is not efficient unless for 15MS4T4G (film). In fact, the results suggested that B9 release occurs almost instantly. In Figure 6B (12MS-5B9), the total release time of the film is around 6.5 h. Decreasing the amount of vitamin (1 wt.%), it is possible to observe that the entire amount is quickly released. This fact might be due to the crystalline structures and, therefore, the unloading of the vitamin. The incorporation of 4% *w*/*w* of plasticizer allows a decrease in the total release time to 40 min (Figure 6C). Decreasing the concentration of MS to 7% *w*/*w*, the total release of 5% *w*/*w* B9 by 7MS4T4G was obtained after approximately 5 h. On the other hand, for the increase in MS concentration to 15 wt.% (Figure 6E), the release time is 78 min. These results corroborate the literature [35]. In fact, small structures with high porosity could be responsible for the faster external diffusion of the vitamin from the polymeric matrices. Carlan et al. reported the complete and very fast (seconds) release of vitamin B1 from modified starch microparticles prepared by a spray-drying process [36]. The suggestion of the quick release of folic acid by modified starch microparticles was also reported [15]. One other report suggested that plasticizers could be used to improve the release characteristics [37]. 

For zein, the best result obtained is for a concentration of 10 wt.% (Figure 7B). In fact, the total amount of B9 is released from the 10Z compact film after 100 min. These results are in accordance with the literature related to zein as an encapsulation agent. Coelho et al. obtained a low B12 release profile with zein microstructures prepared by the electrospinning technique.

Parin et al. reported the slow release rate of folic acid when encapsulated into poly(vinyl alcohol) fibers produced by electrospinning [35]. The slow/fast release might depend on the weak/strong physical bonds between the vitamins and the fiber surface [35].

In the case of Figure 8 and Figure 9, the release profiles depicted were obtained in SGF fluid at 37 °C. As is visualized, the total amount of B9 is rapidly released in SGF fluid with temperature when compared with ethanol (for modified starch) at room temperature. The best release profile obtained was for 12MS4T4G-5B9. In this case, after 20 min, the entire amount of vitamin is released. The same result was not verified for zein structures. Thus, after 75 h, the entire amount is released by the 5Z compact film (Figure 9A).

### 3.3. Antioxidant Activity (AA) of Developed Structures

The B9 antioxidant capacity loaded to the MS and zein structures was evaluated by the ABTS method. The analysis was performed and the results are presented in Table 4. There is an improvement in antioxidant ability for 7MS4T4G-5B9. The opposite is noted in the presence of 1% *w*/*w* of B9 (antioxidant activity of 7.6 mM TE/g). The compact 5Z-5B9 film shows good antioxidant activity. The most well-developed antioxidant structure was seen for the 10Z-1B9 microstructures. The results suggested that compact films with a high surface area present higher antioxidant activity [38,39,40]. Some reports suggested that the polymer used as the encapsulating agent, as well as the pH, has a crucial influence on the microstructures’ antioxidant activity [41,42]. Limpisophon et al. proposed that plasticizers, such as glycerol, might have an impact on the mechanical characteristics of the microstructures [43]. In fact, glycerol will not have a relevant impact on the antioxidant activity of biomolecules but will increase the elasticity of the polymeric matrices.

The microencapsulation of vitamins via a technique such as electrospinning could enhance the stability and antioxidant activity of the compound. It is essential to achieve the optimization of the polymer agent of encapsulation as well as the vitamin concentration. Rodríguez-Sánchez et al. described the physico-chemical characterization of several polymeric systems, including modified starch electrospun fibers, that might be used for food packaging and in biocompatible scaffold materials [44].

## 4. Conclusions

In this work, vitamin B9 was encapsulated into modified starch and zein microstructures by the electrospinning method. Results suggested that the incorporation of tween 80 and glycerol (4 wt.% each) in modified starch might change the gelatinization and, therefore, the polymer crystalline structure, contributing to its hydration and non-rapid recrystallization. 

In particular, 7MS4T4G showed a compact film with 76.30% of 5 wt.% B9 and 150.7 mM TE/g of antioxidant activity. Increasing the concentration to 15 wt%, a film with heterogeneous structures was observed, with a total release time of 78 min and an encapsulation efficiency of 77%. The better structures (film) obtained for 12MS presented around 44% encapsulation efficiency with 5 wt.%. In the case of 5% *w*/*w* zein, a compact film was achieved for 5 wt.% B9 with 93% encapsulation efficiency and antioxidant activity of 129.5 mM TE/g.

In regard to modified starch, its combination with tween 80 and glycerol might be effective for the stabilization of the bioactive ingredients, avoiding the produced crystalline structures. 

The obtained results suggested that the optimization of the process parameters allows the production of electrospun fibers and compact electrosprayed microbeads. These promising vectors have potential to be used in the functional food and nutraceutical industries.

## Figures and Tables

**Figure 1 polymers-14-04337-f001:**
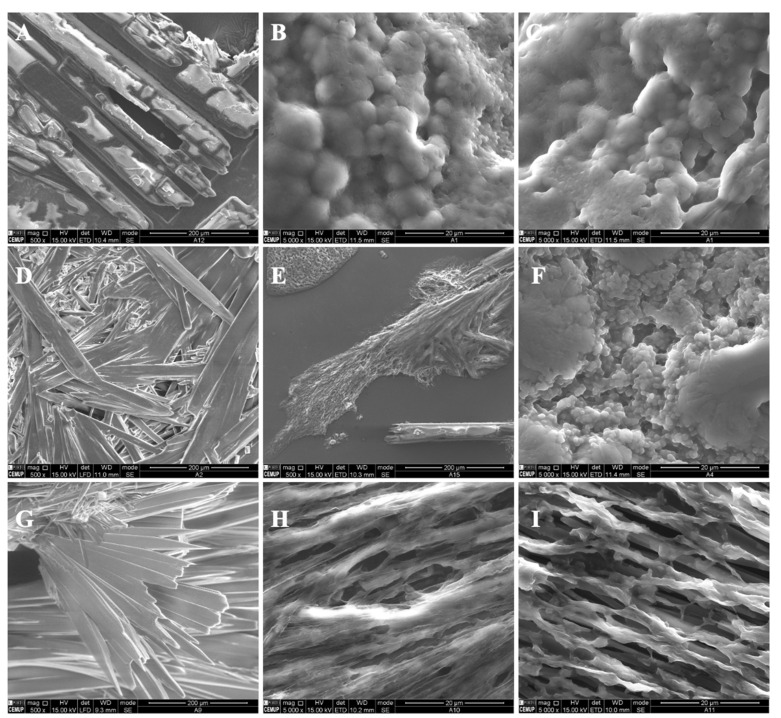
SEM images of electrospun/electrosprayed microstructures produced with (**A**) 7MS, (**B**) 7MS4T4G, (**C**) 7MS8T8G, (**D**) 12MS, (**E**) 12MS4T4G, (**F**) 12MS8T8G, (**G**) 15MS, (**H**) 15MS4T4G and (**I**) 15MS8T8G. Magnification was of 5000×, beam intensity (HV) 15.000 kV, distance between the sample and the lens (WD) less than 12 mm, scale bars of 20 μm.

**Figure 2 polymers-14-04337-f002:**
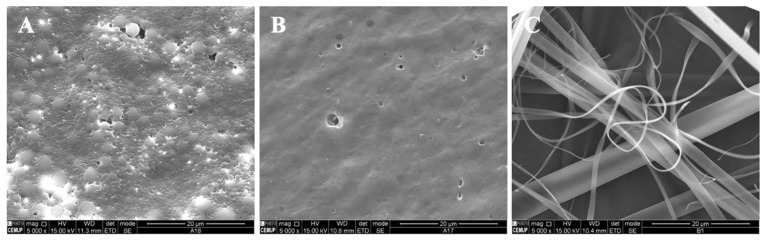
SEM images of electrospun/electrosprayed microstructures produced with (**A**) 5Z, (**B**) 10Z and (**C**) 30Z. Magnification was of 5000×, beam intensity (HV) 15.000 kV, distance between the sample and the lens (WD) less than 12 mm, scale bars of 20 μm.

**Figure 3 polymers-14-04337-f003:**
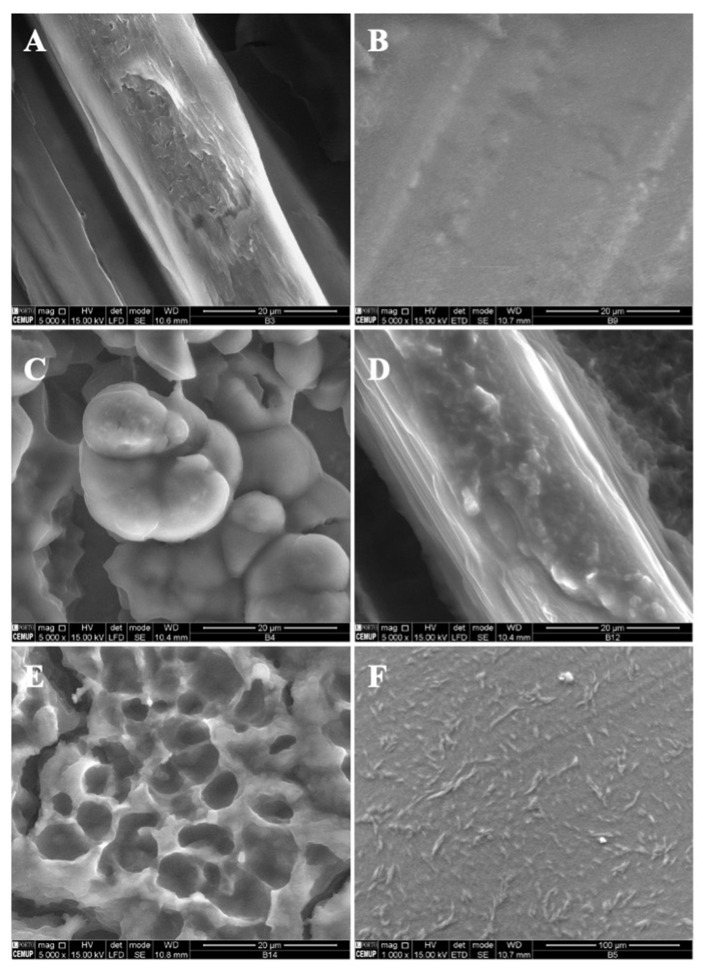
SEM images of electrospun/electrosprayed microstructures produced with (**A**) 12MS-1B9, scale bar is 20 μm; (**B**) 12MS-5B9, scale bar is 20 μm; (**C**) 12MS4T4G-1B9, scale bar is 20 μm; (**D**) 12MS4T4G-5B9, scale bar is 20 μm; (**E**) 12MS8T8G-1B9, scale bar is 20 μm and (**F**) 12MS8T8G-5B9, scale bar is 100 μm. Magnification was of 5000×, beam intensity (HV) 15.000 kV, distance between the sample and the lens (WD) less than 12 mm.

**Figure 4 polymers-14-04337-f004:**
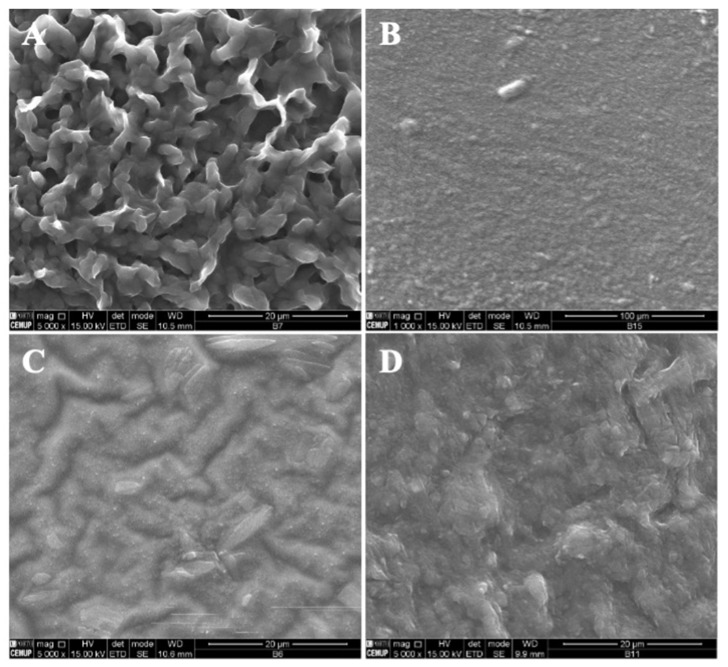
SEM images of electrospun/electrosprayed microstructures produced with (**A**) 7MS4T4G-1B9, scale bar is 20 μm; (**B**) 7MS4T4G-5B9, scale bar is 100 μm; (**C**) 15MS4T4G-1B9, scale bar is 20 μm and (**D**) 15MS4T4G-5B9, scale bar is 20 μm. Magnification was of 5000×, beam intensity (HV) 15.000 kV, distance between the sample and the lens (WD) less than 12 mm.

**Figure 5 polymers-14-04337-f005:**
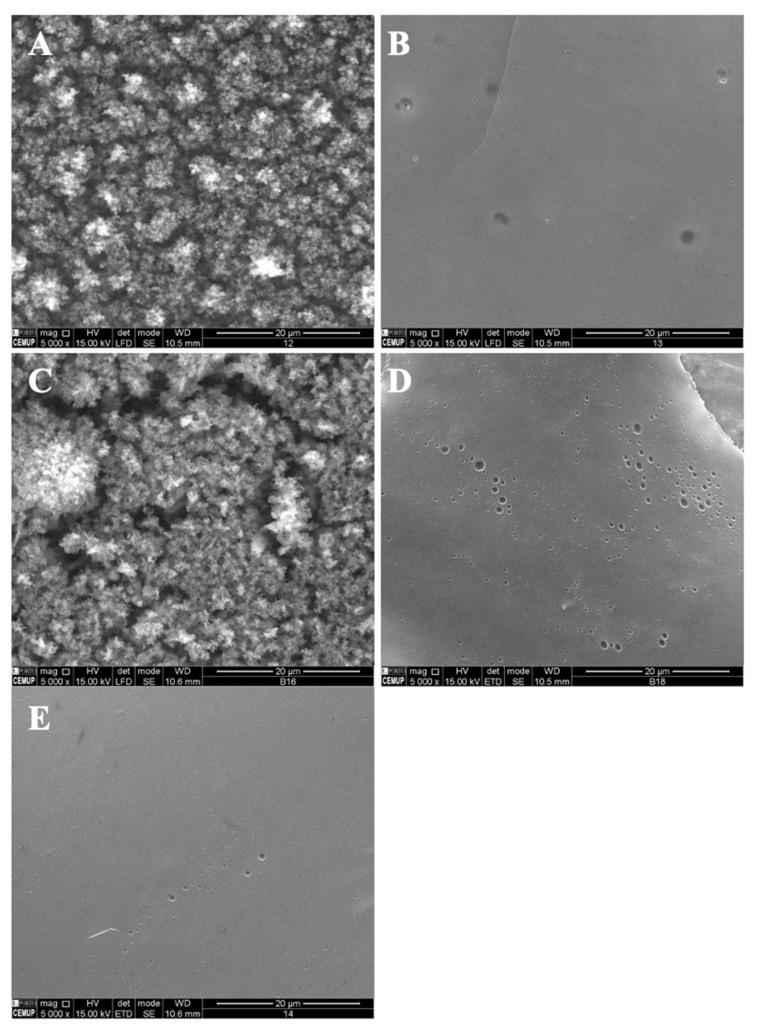
SEM images of electrospun/electrosprayed microstructures produced with (**A**) 5Z-1B9, (**B**) 5Z-5B9, (**C**) 10Z-1B9, (**D**) 10Z-5B9, (**E**) 30Z-1B9. Magnification was of 5000×, beam intensity (HV) 15.000 kV, distance between the sample and the lens (WD) less than 12 mm, scale bars of 20 μm.

**Figure 6 polymers-14-04337-f006:**
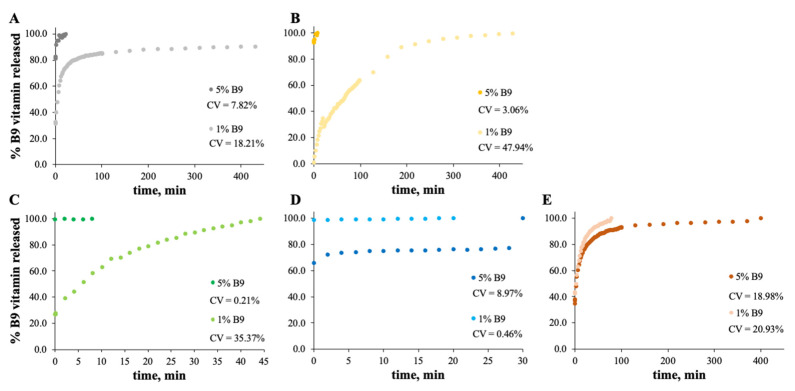
In vitro release profile of vitamin B9 in (**A**) 7MS4T4G, (**B**) 12MS, (**C**) 12MS4T4G, (**D**) 12MS8T8G and (**E**) 15MS4T4G in ethanol. 1% *w*/*w* vitamin B9—clear dots; 5% vitamin B9—dark dots.

**Figure 7 polymers-14-04337-f007:**
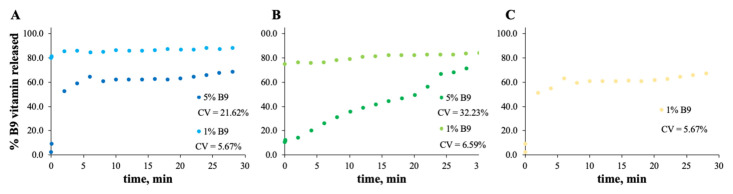
In vitro release profile of vitamin B9 in (**A**) 5Z, (**B**) 10Z and (**C**) 30Z in deionized water. 1% *w*/*w* vitamin B9—clear dots; 5% vitamin B9—dark dots.

**Figure 8 polymers-14-04337-f008:**
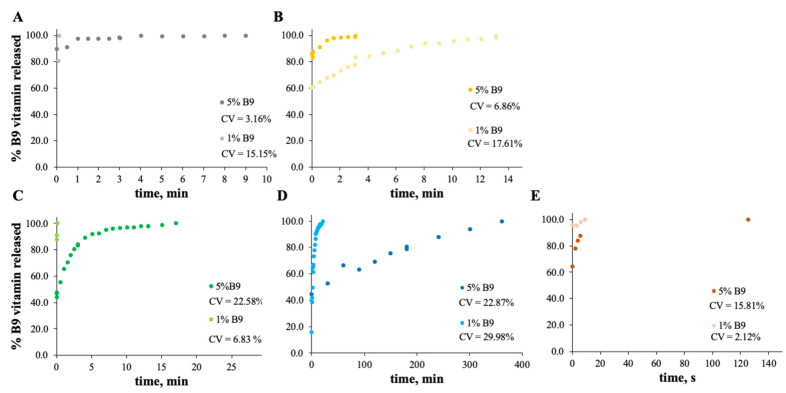
In vitro release profile of vitamin B9 in (**A**) 7MS4T4G, (**B**) 12MS, (**C**) 12MS4T4G, (**D**) 12MS8T8G and (**E**) 15MS4T4G in SGF fluid at 37 °C. 1% *w*/*w* vitamin B9—clear dots; 5% vitamin B9—dark dots.

**Figure 9 polymers-14-04337-f009:**
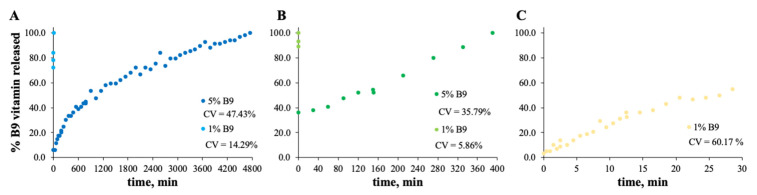
In vitro release profile of vitamin B9 in (**A**) 5Z, (**B**) 10Z and (**C**) 30Z in SGF fluid at 37 °C. 1% *w*/*w* vitamin B9—clear dots; 5% vitamin B9—dark dots.

**Table 1 polymers-14-04337-t001:** Optimized conditions used for fabrication of microstructures by electrospinning technique.

Sample Identification	Sample	Conditions of Method *
7MS	7% *w*/*w* modified starch	0.025 mL/min; 16.29 kV; 5 cm
7MS4T4G	7% *w*/*w* modified starch with 4% *w*/*w* tween 80 and 4% *w*/*w* glycerol	0.05 mL/min; 17.4 kV; 5.5 cm
7MS8T8G	7% *w*/*w* modified starch with 8% *w*/*w* tween 80 and 8% *w*/*w* glycerol	0.025 mL/min; 16.28 kV; 5.5 cm
12MS	12% *w*/*w* modified starch	0.04 mL/min; 15.97 kV; 5 cm
12MS4T4G	12% *w*/*w* modified starch with 4% *w*/*w* tween 80 and 4% *w*/*w* glycerol	0.004 mL/min; 14.45 kV; 10 cm
12MS8T8G	12% *w*/*w* modified starch with 8% *w*/*w* tween 80 and 8% *w*/*w* glycerol	0.025 mL/min; 16.28 kV; 5.5 cm
15MS	15% *w*/*w* modified starch	0.032 mL/min; 20 kV; 10 cm
15MS4T4G	modified starch with 4% *w*/*w* tween 80 and 4% *w*/*w* glycerol	0.1 mL/min; 20 kV; 11.5 cm
15MS8T8G	15% *w*/*w* modified starch with 8% *w*/*w* tween 80 and 8% *w*/*w* glycerol	0.08 mL/min; 17.05 kV; 6 cm
5Z	5% *w*/*w* zein	0.2 mL/h; 20 kV; 7 cm
10Z	10% *w*/*w* zein	0.3 mL/h; 20 kV; 7 cm
30Z	30% *w*/*w* zein	0.3 mL/h; 20 kV; 7 cm

* conditions of electrospinning process: flow rate (mL/h), voltage (kV) and distance (cm).

**Table 2 polymers-14-04337-t002:** Parameters of the analytical method.

Compound	Solvent	Equation	R^2^	LOD (mg/mL)	LOQ (mg/mL)
Vitamin B9	SGF fluid * (37 °C)	y=32.795x+0.0185	0.999	0.0013	0.0041
	Ethanol	y=54.537x+0.0498	0.998	0.0024	0.0074
	Deionized water	y=45.296x+0.0505	0.999	0.0016	0.0048

* Linearity range of vitamin B9 in SGF fluid and H2O: 0.003–0.05 mg/mL; linearity range of vitamin B9 in ethanol: 0.0025–0.05 mg/mL.

**Table 3 polymers-14-04337-t003:** Vitamin B9 encapsulation efficiency (EE) and stability of the produced structures after 2 months.

Samples Identification	Sample	B9 Concentration, mg/mL	EE, %	EE after 2 Months, %	Stability
7MS4T4G-1B9	7MS4T4G-1% *w*/*w* B9	0.002	8.66 ± 0.23	9.20 ± 0.04	0
7MS4T4G-5B9	7MS4T4G-5% *w*/*w* B9	0.033	76.3 ± 0.05	75.6 ± 0.06	0.92
12MS-1B9	12MS-1% *w*/*w* B9	0.021	90.4 ± 0.05	94.37 ± 0.036	0
12MS4T4G-1B9	12MS4T4G-1% *w*/*w* B9	0.008	58.4 ± 0.23	63.06 ± 0.125	0
12MS8T8G-1B9	12MS8T8G-1% *w*/*w* B9	0.005	79.4 ± 0.08	67.03 ± 0.063	15.6
12MS-5B9	12MS-5% *w*/*w* B9	0.018	43.7 ± 0.06	53.55 ± 1.659	0
12MS4T4G-1B9	12MS4T4G-1% *w*/*w* B9	0.056	69.3 ± 0.02	60.76 ± 0.016	12.3
12MS8T8G-5B9	12MS8T8G-5% *w*/*w* B9	0.053	77.3 ± 0.02	75.00 ± 0.114	2.98
15MS4T4G-1B9	15MS4T4G-1% *w*/*w* B9	0.009	23.8 ± 0.47	19.29 ± 0.344	18.9
15MS4T4G-5B9	15MS4T4G-5% *w*/*w* B9	0.038	86.7± 0.10	88.14 ± 0.064	0
5Z-1B9	5Z-1% *w*/*w* B9	0.002	51.7 ± 0.73	48.62 ± 1.990	5.96
5Z-5B9	5Z-5% *w*/*w* B9	0.02	92.7 ± 2.66	86.24 ± 1.390	6.97
10Z-1B9	10Z-1% *w*/*w* B9	0.003	56.5 ± 0.22	55.05 ± 0.143	2.57
10Z-5B9	10Z-5% *w*/*w* B9	0.007	95.2 ± 0.48	92.36 ± 0.781	2.98
30Z-1B9	30Z-1% *w*/*w* B9	0.002	82.4 ± 0.68	83.45 ± 2.450	0

**Table 4 polymers-14-04337-t004:** Antioxidant activity (AA) of fabricated microstructures (ABTS assay).

Sample Identification	Antioxidant Activity, mM TE/g Sample, AA	
	Control	Sample
7MS4T4G-1B9	56.15 ± 1.700	7.6 ± 0.1
7MS4T4G-5B9	44.40 ± 2.017	150.7 ± 0.285
12MS-1B9	75.65 ± 1.877	93.57 ± 0.196
12MS4T4G-1B9	164.1 ± 0.250	13.8 ± 0.10
12MS8T8G-1B9	144.64 ± 6.5	61.76 ± 0.439
12MS-5B9	17.96 ± 9.5	121.8 ± 1.879
12MS4T4G-1B9	35.5 ± 0.05	30.15 ± 0.200
12MS8T8G-5B9	8.71 ± 0.10	10.5 ± 0.100
15MS4T4G-1B9	101.3 ± 1.356	71.6 ± 0.09
15MS4T4G-5B9	44.58 ± 0.352	75.6 ± 0.08
5Z-1B9	428.9 ± 8.264	69.6 ± 5.24
5Z-5B9	303.0 ± 3.600	129.5 ± 0.023
10Z-1B9	67.65 ± 2.900	302.18 ± 0.320
10Z-5B9	157.94 ± 2.812	131.6 ± 0.220
30Z-1B9	142.42 ± 0.300	80.02 ± 0.100

## Data Availability

Data available on request due to restrictions eg privacy or ethical. The data presented in this study are available on request from the corresponding author.

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
