# Peer review of "Electrospinning of Microstructures Incorporated with Vitamin B9 for Food Application: Characteristics and Bioactivities"

_polymers, 2022, doi:10.3390/polym14204337_

Round 1

Reviewer 1 Report

General comments

The present manuscript is focused on the encapsulation of B9 vitamin within modified starch with two plasticizers and zein based  systems.

However, the paper quality is very low with respect to the high standards of Polymers. The originality is not clear, since previously other papers about the same topic, using the same materials, were published.

The manuscript is very confusing. It is not clear why both modified starch and zein based structures were proposed. The Abstract is not clear and has to be completely rewritten, the Materials and Methods section is not complete and has to be improved in order to make all the methods reproducible.

The Results and Discussion section has to be deeply extended, since the Authors only described the acquired data, but they did not justify and discuss them. They failed in comparing their achievements with the Literature.

Moreover, they obtained neither fibers nor single particles….the electrospinning and electrospraying processes need to be set up in order to obtain uniform, homogeneous, defect-free structures.

An editing of English language and style is required. Some specific remarks and suggestions are reported below point by point.

Abstract

- The Abstract section is not clear and has to be completely rewritten.

- A contextualisation has to be added as incipit.

- The nomenclature “12MS-15 1B9 film, 5Z-5B9 compact film, 10Z-5B9 film and 15MS4T4G-5B9 film” cannot be used without having been defined before.

Keywords

-        The chosen keywords (i.e. Electrospinning; Zein protein; Modified starch; Plasticizers; Gelatinization; Vitamin B9) do not completely cover the manuscript content and, thus, further ones have to be added.

-        Moreover, the keywords have to be reported in a more logical order (materials, synthesis/processing, characterisations, properties, application).

1. Introduction

- The Introduction section should be better organized, since in some points the Authors anticipated the aim of their work, then they reported some literature results about the topic, without a logical order.

- The incipit “Electrospun/electrosprayed microstructures are systems able to overcome the limitations associated to the presented vitamins in functional food and nutraceutical products. The low bioavailability, stability and half-life time are the main drawbacks of these ingredients” needs to be corroborated with suitable references.

- The phrase “The focus of this study is the vitamin encapsulation in polymeric microstructures prepared by electrospinning method.” sounds out of place. Please remove it in tis point of the Introduction section.

- The Authors stated that “Electropinning and electropraying are recent and efficient electrohydrodynamic techniques to produce low-cost and versatile polymeric microstructures (fibres, film and/or particles)”, but they are very old techniques.

The originality and added value to the scientific community of the present research paper has to be evidenced at the end of the Introduction section. It seems that similar systems have been already produced in previous papers.

- It is strongly recommended to report a brief list of the performed characterisations.

2. Materials and Methods

2.1. Materials

- More details, such as the purity of some reagents and solvents, as well as the molecular weight for the used polymers, have to be added.

2.2. Preparation of the zein and modified starch solutions

- The ethanol: water ratio has to be specified.

- The following phrase “Different zein concentrations – 5% w/w, 10% w/w and 30% w/w in 70%, 70% and 60% (w/w, weight of solvent/(weight of solvent + weight of water)) ethanol, respectively, were prepared (table 1).” is not clear. Please, rewrite it.

- For the modified starch solutions, more details have to be added. The chosen MS concentrations have to be justified, as well as those of the glycerol and Tween 80.

2.3. Production of microstructures by electrospinning

- The diameter of the used needles have to be specified.

- How was the temperature set at 22 °C? was the relative humidity monitored?

2.1. Characterization of microstructures

- How were the samples prepared for SEM observation? Please, add details.

2.2. In vitro release studies

-It is not clear how B9 vitamin was encapsulated and the initial concentration. These details have to be reported in the paragraph about the preparation of the solution.

- More details about how the calibration curve, coefficient of variation, limit of detection (LOD) and limit of quatification (LOQ) were determined for vitamin B9 have to be added.

 - The Authors stated that “The release studies were performed adding microparticles to the release medium, incubated with constant magnetic stirring.”, but they produced both fibers and particles. Moreover, they have to specify the particles (fibers)/medium ratio.

- The paragraph “2.4. Statistical analysis” has to be properly expanded.

3. Results

3.1. Characterization of produced structures in terms of morphology and size and encapsulation efficiency (EE) - As – The used B9 vitamin concentrations have to be reported in the Experimental section and not in the Results one.

- The following considerations “These results corroborate the literature suggesting that the incorporation with plasticizers – glycerol and tween 80 – will improve tensile strength in the MS which does not recrystalise quickly. The best structures to use for food applications are produced with 4% w/w of plasticizers. These circumstances might be a benefit to the better mechanical characteristics of the produced microstructures” do not sound appropriate, since the authors did not investigate the mechanical properties of the produced structures.

3.2. In vitro release experiments

- The reported results have not only to be described, but also discussed and justified, comparing them with the literature.

3.3. Antioxidant activity (AA) of developed structures

- As evidenced before, the acquired data have to be discussed and justified.

Author Response

Response to the reviewers

We would like to thank the comments of the reviewers on the manuscript “Electrospinning of microstructures incorporated with Vitamin B9 for food application: Characteristics and bioactivities “.

The manuscript has been modified according to the suggestions of the reviewers.

Reviewer 1

Comments and Suggestions for Authors

General comments

The present manuscript is focused on the encapsulation of B9 vitamin within modified starch with two plasticizers and zein based  systems.

However, the paper quality is very low with respect to the high standards of Polymers. The originality is not clear, since previously other papers about the same topic, using the same materials, were published.

  • The authors respect the opinion of the reviewer. The paper was improved and reformulated. The originality was clarified.

The manuscript is very confusing. It is not clear why both modified starch and zein based structures were proposed. The Abstract is not clear and has to be completely rewritten, the Materials and Methods section is not complete and has to be improved in order to make all the methods reproducible.

  • Suggestion accepted. New sentences were included in the “Introduction” and “Materials and methods” sections. The abstract was also revised.

The Results and Discussion section has to be deeply extended, since the Authors only described the acquired data, but they did not justify and discuss them. They failed in comparing their achievements with the Literature.

  • Thank you for the suggestion. New text was inserted in the Results and Discussion section.

Moreover, they obtained neither fibers nor single particles….the electrospinning and electrospraying processes need to be set up in order to obtain uniform, homogeneous, defect-free structures.

  • Thank you for the comment. By electrospinning and electrospraying processes 3 different types of structures can be obtained: films, fibres and particles. In ideal/optimized conditions a specific type of structures is obtained, as it was described by the authors in different papers (Coelho et al 2021 - A new approach to the production of zein microstructures with vitamin B12, by electrospinning and spray drying techniques; Coelho et al 2022 - Optimization of electrospinning parameters for the production of zein microstructures for food and biomedical applications). However, there are several reasons to obtain different results, namely the incorporation of the two plasticizers.

An editing of English language and style is required. Some specific remarks and suggestions are reported below point by point.

  • Thank you for the suggestion. The entire text was revised.

Abstract

- The Abstract section is not clear and has to be completely rewritten.

- A contextualisation has to be added as incipit.

- The nomenclature “12MS-15 1B9 film, 5Z-5B9 compact film, 10Z-5B9 film and 15MS4T4G-5B9 film” cannot be used without having been defined before.

- Suggestion accepted. Abstract was rewritten. Please see the new text.

Keywords

-        The chosen keywords (i.e. Electrospinning; Zein protein; Modified starch; Plasticizers; Gelatinization; Vitamin B9) do not completely cover the manuscript content and, thus, further ones have to be added.

-        Moreover, the keywords have to be reported in a more logical order (materials, synthesis/processing, characterisations, properties, application).

- Thank you for the suggestion. New keywords were selected.

  1. Introduction

- The Introduction section should be better organized, since in some points the Authors anticipated the aim of their work, then they reported some literature results about the topic, without a logical order.

- The incipit “Electrospun/electrosprayed microstructures are systems able to overcome the limitations associated to the presented vitamins in functional food and nutraceutical products. The low bioavailability, stability and half-life time are the main drawbacks of these ingredients” needs to be corroborated with suitable references.

- The phrase “The focus of this study is the vitamin encapsulation in polymeric microstructures prepared by electrospinning method.” sounds out of place. Please remove it in tis point of the Introduction section.

- The Authors stated that “Electropinning and electropraying are recent and efficient electrohydrodynamic techniques to produce low-cost and versatile polymeric microstructures (fibres, film and/or particles)”, but they are very old techniques.

The originality and added value to the scientific community of the present research paper has to be evidenced at the end of the Introduction section. It seems that similar systems have been already produced in previous papers.

- It is strongly recommended to report a brief list of the performed characterisations.

- The suggestions were accepted. The modifications are included in the revised version of the manuscript.

  1. Materials and Methods

2.1. Materials

- More details, such as the purity of some reagents and solvents, as well as the molecular weight for the used polymers, have to be added.

-  Additional information was added.

2.2. Preparation of the zein and modified starch solutions

- The ethanol: water ratio has to be specified.

- The following phrase “Different zein concentrations – 5% w/w, 10% w/w and 30% w/w in 70%, 70% and 60% (w/w, weight of solvent/(weight of solvent + weight of water)) ethanol, respectively, were prepared (table 1).” is not clear. Please, rewrite it.

- For the modified starch solutions, more details have to be added. The chosen MS concentrations have to be justified, as well as those of the glycerol and Tween 80.

- Suggestions accepted. Some sentences were rewritten. Please see the Materials and Methods section.

2.3. Production of microstructures by electrospinning

- The diameter of the used needles have to be specified.

- How was the temperature set at 22 °C? was the relative humidity monitored?

- Suggestion accepted. The diameter of the needle was specified in this section. The temperature set at 22°C. The humidity was not monitored, but the authors recognize that it is an important factor.

2.1. Characterization of microstructures

- How were the samples prepared for SEM observation? Please, add details.

- New details were inserted in this section.

2.2. In vitro release studies

-It is not clear how B9 vitamin was encapsulated and the initial concentration. These details have to be reported in the paragraph about the preparation of the solution.

- More details about how the calibration curve, coefficient of variation, limit of detection (LOD) and limit of quatification (LOQ) were determined for vitamin B9 have to be added.

- New text was inserted in this subsection.

 - The Authors stated that “The release studies were performed adding microparticles to the release medium, incubated with constant magnetic stirring.”, but they produced both fibers and particles. Moreover, they have to specify the particles (fibers)/medium ratio.

  • Thank you for your comment. The text was modified.

- The paragraph “2.4. Statistical analysis” has to be properly expanded.

              - Suggestion accepted. New sentence was added to this subsection.

  1. Results

3.1. Characterization of produced structures in terms of morphology and size and encapsulation efficiency (EE) - As – The used B9 vitamin concentrations have to be reported in the Experimental section and not in the Results one.

- Suggestion accepted. Please see the new text included in the Results section.

- The following considerations “These results corroborate the literature suggesting that the incorporation with plasticizers – glycerol and tween 80 – will improve tensile strength in the MS which does not recrystalise quickly. The best structures to use for food applications are produced with 4% w/w of plasticizers. These circumstances might be a benefit to the better mechanical characteristics of the produced microstructures” do not sound appropriate, since the authors did not investigate the mechanical properties of the produced structures.

- New sentence was included in the section.

3.2. In vitro release experiments

- The reported results have not only to be described, but also discussed and justified, comparing them with the literature.

- Suggestion accepted. Please see the new text.

3.3. Antioxidant activity (AA) of developed structures

- As evidenced before, the acquired data have to be discussed and justified.

- Thank you for the suggestion. Please see the new sentences included in this section that corroborate the literature.

Reviewer 2 Report

Dear authors,

Article structure is prepared according to journal Polymers

The topic is in the scope of the journal Polymers. Article presents interesting and current topic for food applications - the microstructure electrospinning with the incorporation of vitamin.

Here are my comments:

1)     Abstract: is clearly presented.

2)     Introduction should include more background regarding electrospinning and incorporation of vitamins, since the authors published review nearly to the topic of this manuscript. The references are not written correctly. This should be corrected. Please check the authors instructions (should not be overwritten). The aim and the goal of the research are clearly presented.

3)     Line 36: correct the reference number 34. The same mistake is at line 46 etc. References are not listed in order.

4)     Materials and methods: add the country at each material provided.

5)     Chapter 2.2.: what were the stirring conditions?

6)     Table 3: how was stability of B9 vitamin encapsulation measured/determined? Add this in the materials and method chapter.

7)     Chapter 3.3 Antioxidant activity: add the description of this method to the chapter Materials and methods.

8)     The conclusion: is clearly presented.

9)     References: some are not correctly written.  As mentioned before the number of the cited references should be in order in the whole manuscript.  

10)  Many typos are presented in the manuscript.

Author Response

Response to the reviewers

We would like to thank the comments of the reviewers on the manuscript “Electrospinning of microstructures incorporated with Vitamin B9 for food application: Characteristics and bioactivities “.

The manuscript has been modified according to the suggestions of the reviewers.

Reviewer 2

Comments and Suggestions for Authors

Dear authors,

Article structure is prepared according to journal Polymers

The topic is in the scope of the journal Polymers. Article presents interesting and current topic for food applications - the microstructure electrospinning with the incorporation of vitamin.

  • We would like to thank the comments of the reviewers.

Here are my comments:

  • Abstract: is clearly presented.

- Thank you.

2)     Introduction should include more background regarding electrospinning and incorporation of vitamins, since the authors published review nearly to the topic of this manuscript. The references are not written correctly. This should be corrected. Please check the authors instructions (should not be overwritten). The aim and the goal of the research are clearly presented.

- The introduction was reformulated. New text was inserted in Introduction section.

3)     Line 36: correct the reference number 34. The same mistake is at line 46 etc. References are not listed in order.

- The correction was made.

4)     Materials and methods: add the country at each material provided.

- New information was added.

5)     Chapter 2.2.: what were the stirring conditions?

- The information was added to the paper (300 rpm).

6)     Table 3: how was stability of B9 vitamin encapsulation measured/determined? Add this in the materials and method chapter.

7)     Chapter 3.3 Antioxidant activity: add the description of this method to the chapter Materials and methods.

- In vitro stability assay (section 2.7) and In vitro antioxidant activity experiment (2.8) were included in Materials and Methods section.

8)     The conclusion: is clearly presented.

- Thank you.

9)     References: some are not correctly written.  As mentioned before the number of the cited references should be in order in the whole manuscript.  

- References section was improved and revised.

10)  Many typos are presented in the manuscript.

- The paper was improved and revised. Several typos were corrected.

Reviewer 3 Report

The paper ‘Electrospinning of microstructures incorporated with Vitamin B9 for food application: Characteristics and bioactivities’ by Coelho et al. investigated two approaches to encapsulating vitamin B9. The combination of modified starch with two plasticizers and zein were the encapsulating agents used to produce microstructures by electrospinning technique to boost the B9 antioxidant capacity.

Also, the influence of plasticizers – tween 80 and glycerol–on the gelatinization of the modified starch and, therefore, on the developed crystalline structures was investigated. The work is well-written, properly designed, and discussed. However, the following points need to be addressed:

1.       Amend Table 2 likewise Table 1, use border and shading.

2.       The statistical analysis tools, number of replicas (where relevant), and the used software is missed in the statistical analysis part

3.       More detail about the SEM is needed, e.g. coating, voltage, etc.

4.       Figure 3 F) 12MS8T8G-5B9, and also Figure 4 (b and c) the pictures are blurry, could you provide a better resolution?

5.       Line 211 (beam intensity (HV)15.000 kV), it seems that this intensity (15 kV) is high for the electrospun biodegradable materials.

6.       In Tables 3 and 4, could you show the significant difference letters to distinguish the treatment similarities/differences?

7.       Figures 6-9, a better visual quality of the presented results is expected, making them more eye-appealing by using fancy colorful figures.

8.       Have you done any replicas for the results presented in Figures 6-9? If so, provide the error bars on them.

Author Response

Response to the reviewers

We would like to thank the comments of the reviewers on the manuscript “Electrospinning of microstructures incorporated with Vitamin B9 for food application: Characteristics and bioactivities “.

The manuscript has been modified according to the suggestions of the reviewers.

Reviewer 3

Comments and Suggestions for Authors

The paper ‘Electrospinning of microstructures incorporated with Vitamin B9 for food application: Characteristics and bioactivities’ by Coelho et al. investigated two approaches to encapsulating vitamin B9. The combination of modified starch with two plasticizers and zein were the encapsulating agents used to produce microstructures by electrospinning technique to boost the B9 antioxidant capacity.

Also, the influence of plasticizers – tween 80 and glycerol–on the gelatinization of the modified starch and, therefore, on the developed crystalline structures was investigated. The work is well-written, properly designed, and discussed. However, the following points need to be addressed:

- Thank you for your comments. The paper was improved based in your comments.

  1. Amend Table 2 likewise Table 1, use border and shading.

              - Table 2 was modified.

  1. The statistical analysis tools, number of replicas (where relevant), and the used software is missed in the statistical analysis part

- Statistical analysis section was improved.

  1. More detail about the SEM is needed, e.g. coating, voltage, etc.

              - SEM details were included in 2.4. section.

  1. Figure 3 F) 12MS8T8G-5B9, and also Figure 4 (b and c) the pictures are blurry, could you provide a better resolution?

              - Figures 3 and 4 are provided with better resolution. Please see the images.

  1. Line 211 (beam intensity (HV)15.000 kV), it seems that this intensity (15 kV) is high for the electrospun biodegradable materials.

                     - The SEM images containing the voltage value (!5.00 KV). This value it can appear high but it was the value that was used in electrospun biodegradable materials, without problems.

  1. In Tables 3 and 4, could you show the significant difference letters to distinguish the treatment similarities/differences?

                      - It was not possible in a useful time to add this information, related to the significance test.

  1. Figures 6-9, a better visual quality of the presented results is expected, making them more eye-appealing by using fancy colorful figures.

              - Figures 6-9 were modified using colours. Coefficient of variation (CV%) were included in the graphs.

  1. Have you done any replicas for the results presented in Figures 6-9? If so, provide the error bars on them.

- The error bars would turn the results difficult to observe, for this reason information related to the coefficient variation that affect the results was added in the legend of the figures.

Round 2

Reviewer 1 Report

General comments

I confirm my previous decision, since the Authors did not significantly and properly revised the paper, but most of the requested corrections have not been applied, as reported below point by point.

The paper quality has not been improved

Abstract

- The Abstract section continues to be not clear. The Authors immediately report the main results, without having explained how they reached the study objective.

- The incipit “Food industry has been increasing and new vectors have been constantly investigated aiming versatile systems with good physico-chemical characteristics, low-cost production and high stability.” is too generic and has to be correlated to the use of vitamins.

1. Introduction

- The sentence “ The focus of this study is the vitamin encapsulation in prepared polymeric microstructures by electrospinning method” is out of place. The work aim has to be reported at the end of the Introduction section, after the state of the art about the paper topic.

- The phrase “The focus of this study is the vitamin encapsulation in polymeric microstructures prepared by electrospinning method.” sounds out of place. Please remove it in tis point of the Introduction section.

- Even if already requested in the previous review, the originality and added value to the scientific community of the present research paper has not been highlighted yet at the end of the Introduction section.

- The conclusion “The results suggested the optimization of some parameters that allowed the production of electrospun fibres and compact electrosprayed microbeads that might be promising vectors for food and biomedical applications” does not sound appropriate for the Introductions ection, but it is more proper for the Conclusions one.

2. Materials and Methods

2.1. Materials

- As requested in the previous review, more details, such as the purity of some reagents and solvents, as well as the molecular weight for the used polymers, have to be added.

2.2. Preparation of the zein and modified starch solutions

- The ethanol: water ratio has to be specified, as previously requested.

- For the modified starch solutions, more details have to be added. The chosen MS concentrations have to be justified, as well as those of the glycerol and Tween 80, as previously requested.

2.2. In vitro release studies

- More details about how the calibration curve, coefficient of variation, limit of detection (LOD) and limit of quatification (LOQ) were determined for vitamin B9 have to be added, as previously requested.

 - The Authors stated that “The release studies were performed adding microparticles to the release medium, incubated with constant magnetic stirring.”, but they produced both fibers and particles.

- Moreover, they have to specify the particles (fibers)/medium ratio, as already requested in the previous review.

2.4. Characterization of microstructures

For the SEM, the time and applied voltage/current for the gold coating have to be specified.

3. Results

3.2. In vitro release experiments

- Concerning the following revision requested in the previous review, i.e. “The reported results have not only to be described, but also discussed and justified, comparing them with the literature”, the Authors added only 2 lines, not significantly improving this paragraph.

3.3. Antioxidant activity (AA) of developed structures

- Similarly to the previous paragraph, the Authors added two lines, but, as evidenced in the previous review, the acquired data have to be discussed and justified.

Author Response

Response to the reviewers

We would like to thank the comments of the reviewers on the manuscript “Electrospinning of microstructures incorporated with Vitamin B9 for food application: Characteristics and bioactivities “.

The manuscript has been modified according to the suggestions of the reviewers.

Reviewer 1

Comments and Suggestions for Authors

General comments

I confirm my previous decision, since the Authors did not significantly and properly revised the paper, but most of the requested corrections have not been applied, as reported below point by point.

The paper quality has not been improved

Answer: The manuscript has been modified according to the suggestions of all the reviewers.

Abstract

- The Abstract section continues to be not clear. The Authors immediately report the main results, without having explained how they reached the study objective.

Answer: Thank you for the suggestion. The abstract was modified. The main experiments of this study were included in the abstract.

- The incipit “Food industry has been increasing and new vectors have been constantly investigated aiming versatile systems with good physico-chemical characteristics, low-cost production and high stability.” is too generic and has to be correlated to the use of vitamins.

Answer: Suggestion accepted. The sentence was reformulated.

  1. Introduction

- The sentence “ The focus of this study is the vitamin encapsulation in prepared polymeric microstructures by electrospinning method” is out of place. The work aim has to be reported at the end of the Introduction section, after the state of the art about the paper topic.

Answer: The sentence was removed.

- The phrase “The focus of this study is the vitamin encapsulation in polymeric microstructures prepared by electrospinning method.” sounds out of place. Please remove it in tis point of the Introduction section.

Answer: The sentence was removed from the Introduction section.

- Even if already requested in the previous review, the originality and added value to the scientific community of the present research paper has not been highlighted yet at the end of the Introduction section.

Answer: Additional information was inserted in the Introduction section.

- The conclusion “The results suggested the optimization of some parameters that allowed the production of electrospun fibres and compact electrosprayed microbeads that might be promising vectors for food and biomedical applications” does not sound appropriate for the Introductions section, but it is more proper for the Conclusions one.

Answer: Thank you for the suggestion. The sentence was modified and included in the Conclusion section.

  1. Materials and Methods

2.1. Materials

- As requested in the previous review, more details, such as the purity of some reagents and solvents, as well as the molecular weight for the used polymers, have to be added.

Answer: The available information about the reagents was added in the document. There are some details that are not specified in the commercial information sheets of the reagents.

2.2. Preparation of the zein and modified starch solutions

- The ethanol: water ratio has to be specified, as previously requested.

Answer: Thank you for the suggestion. The preparation of the solutions and the ratio of ethanol are presented in subsection 2.2.

- For the modified starch solutions, more details have to be added. The chosen MS concentrations have to be justified, as well as those of the glycerol and Tween 80, as previously requested.

Answer: Suggestion accepted. The preparation of the starch solution was clarified in the text. Preliminary studies were performed to optimize the process of modified starch (MS) microstructures fabrication and are visualized in table 1. This optimization was executed based on literature:

Kong et al. reported that 15% w/w starch fibres can be developed as promising vectors for food and biomedical products [1]. Based on this research and our goal, the optimization of electrospinning parameters initiated with 15 % w/w MS concentration. Different concentrations below 15% (7% w/w and 12% w/w) were also evaluated, aiming the fabrication of microstructures with small diameters.

Several reports suggested the combination of plasticizers and modified starch to improve mechanical characteristics of the structures [2][3]. Therefore, this study had one other goal that consisted in the combination of two plasticizers (glycerol and tween 80)[4][5] with MS in order to enhance the vitamin B9 encapsulation efficiency [6].

Please see the new text in the subsection 2.2.

  1. Kong, L.; Ziegler, G.R. Fabrication of pure starch fibers by electrospinning. Food Hydrocoll. 2014, 36, 20–25, doi:https://doi.org/10.1016/j.foodhyd.2013.08.021.
  2. Temesgen, S.; Rennert, M.; Tesfaye, T.; Nase, M. Review on Spinning of Biopolymer Fibers from Starch. Polym. 2021, 13.
  3. Curvelo, A.A.S.; de Carvalho, A.J.F.; Agnelli, J.A.M. Thermoplastic starch–cellulosic fibers composites: preliminary results. Carbohydr. Polym. 2001, 45, 183–188, doi:https://doi.org/10.1016/S0144-8617(00)00314-3.
  4. Tarique, J.; Sapuan, S.M.; Khalina, A. Effect of glycerol plasticizer loading on the physical, mechanical, thermal, and barrier properties of arrowroot (Maranta arundinacea) starch biopolymers. Sci. Rep. 2021, 11, 13900, doi:10.1038/s41598-021-93094-y.
  5. Ivanič, F.; Kováčová, M.; Chodák, I. The effect of plasticizer selection on properties of blends poly(butylene adipate-co-terephthalate) with thermoplastic starch. Eur. Polym. J. 2019, 116, 99–105, doi:https://doi.org/10.1016/j.eurpolymj.2019.03.042.
  6. Ledezma-Oblea, J.G.; morales sanchez, E.; Marcela, G.; Figueroa, J.; Gaona-Sánchez, V.A. Corn starch nanofilaments obtained by electrospinning. Rev. Mex. Ing. química 2015, 14, 497–502.

2.2. In vitro release studies

- More details about how the calibration curve, coefficient of variation, limit of detection (LOD) and limit of quatification (LOQ) were determined for vitamin B9 have to be added, as previously requested.

Answer: Suggestion accepted. New text was inserted in subsection 2.5.

 - The Authors stated that “The release studies were performed adding microparticles to the release medium, incubated with constant magnetic stirring.”, but they produced both fibers and particles.

Answer: Thank you for the suggestion. The denomination of the structures was changed on the sentence. Please see subsection 2.5.

- Moreover, they have to specify the particles (fibers)/medium ratio, as already requested in the previous review.

Answer: Additional information was added.

2.4. Characterization of microstructures

For the SEM, the time and applied voltage/current for the gold coating have to be specified.

Answer: The coating process takes 15 minutes. The voltage used it was not available.

  1. Results

3.2. In vitro release experiments

- Concerning the following revision requested in the previous review, i.e. “The reported results have not only to be described, but also discussed and justified, comparing them with the literature”, the Authors added only 2 lines, not significantly improving this paragraph.

Answer: The discussion was improved. New text was inserted in the subsection 3.2.

3.3. Antioxidant activity (AA) of developed structures

- Similarly to the previous paragraph, the Authors added two lines, but, as evidenced in the previous review, the acquired data have to be discussed and justified.

Answer: New text was added in the subsection 3.3.

Reviewer 2 Report

Dear authors,

You have corrected the manuscript as suggested. It can be now accepted in the present form.

Author Response

Answer: Thank you for your comment

Round 3

Reviewer 1 Report

The paper can be accepted in the current version.